# Processing, Microstructure and Mechanical Properties of TiB₂-MoSi₂-C Ceramics

**Maria Sajdak** [1] , **Kamil Kornaus** [1,*], **Dariusz Zientara** [1] , **Norbert Moskała** [1] , **Sebastian Komarek** [1] , **Kinga Momot** [2], **Edmund Golis** [3], **Łukasz Zych** [1] **and Agnieszka Gubernat** [1]

[1] Faculty of Materials Science and Ceramics, Department of Ceramics and Refractories, AGH University of Krakow, al. Adama Mickiewicza 30, 30-059 Kraków, Poland; msajdak@agh.edu.pl (M.S.); zientara@agh.edu.pl (D.Z.); nmos1@agh.edu.pl (N.M.); seko@agh.edu.pl (S.K.); lzych@agh.edu.pl (Ł.Z.); gubernat@agh.edu.pl (A.G.)

[2] Łukasiewicz Research Network, Krakow Institute of Technology, Centre of Materials and Manufacturing Technologies, 73 Zakopiańska Str., 30-418 Kraków, Poland; kinga.momot@kit.lukasiewicz.gov.pl

[3] Faculty of Science and Technology, Department of Experimental and Applied Physics, Jan Dlugosz University in Czestochowa, al. Armii Krajowej 13/15, 42-200 Częstochowa, Poland; e.golis@ujd.edu.pl

[*] Correspondence: kornaus@agh.edu.pl

**Abstract:** Titanium boride (TiB₂) is a material classified as an ultra-high-temperature ceramic. The TiB₂ structure is dominated by covalent bonds, which gives the materials based on TiB₂ very good mechanical and thermal properties, making them difficult to sinter at the same time. Obtaining dense TiB₂ polycrystals requires a chemical or physical sintering activation. Carbon and molybdenum disilicide (MoSi₂) were chosen as sintering activation additives. Three series of samples were made, the first one with carbon additives: 0 to 4 wt.%; the second used 2.5, 5 and 10 wt.% MoSi₂; and the third with both additions of 2 wt.% carbon and 2.5, 5 and 10 wt.% MoSi₂. On the basis of the dilatometric sintering analysis, all additives were found to have a favourable effect on the sinterability of TiB₂, and it was determined that sintering TiB₂ with the addition of carbon can be carried at 2100 °C and with MoSi₂ and both additives at 1800 °C. The polycrystals were sintered using the hot-pressing technique. On the basis of the studies conducted in this work, it was found that the addition of 1 wt.% of carbon allows single-phase TiB₂ polycrystals of high density (>90%) to be obtained. The minimum MoSi₂ addition, required to obtain dense sinters with a cermet-like microstructure, was 5 wt.%. High density was also achieved by the materials containing both additives. The samples with higher MoSi₂ content, i.e., 5 and 10%, showed densities close to 100%. The mechanical properties, such as Young's modulus, hardness and fracture toughness ($K_{Ic}$), of the polycrystals and composites were similar for samples with densities exceeding 95%. The Vickers hardness was 23 to 27 GPa, fracture toughness ($K_{IC}$) was 4 to 6 MPa·m$^{0.5}$ and the Young's modulus was 480 to 540 GPa. The resulting TiB₂-based materials showed potential in high-temperature applications.

**Keywords:** TiB₂; MoSi₂; carbon; UHTC; composites; mechanical properties





## 1. Introduction

Ceramic materials classified as ultra-high-temperature ceramics (UHTCs) are characterised by a high melting point, good mechanical properties at high temperature and high oxidation resistance. As they are increasingly used, more demands are put on them [1,2]. The group of materials classified as UHTCs includes metal borides in the fourth group of the periodic table of chemical elements, i.e., TiB₂, ZrB₂ and HfB₂. These borides have very high melting points (approx. 3000 °C), good thermal and electrical conductivity, high hardness, good mechanical properties and oxidation resistance. These valuable properties of AlB₂-type borides are the result of dominant covalent bonds present in their structure, which, on the other hand, has a negative effect on the sinterability of boride ceramics [1–4].

Improved sinterability of boride ceramics is achieved by using sintering additives (sintering activators), such as nitrides TiN, AlN, $Si_3N_4$ or HfN [2,5–9]; carbides TaC, SiC, $B_4C$ [2,10–15]; silicides $MoSi_2$, $TiSi_2$, $TaSi_2$ [4,16–23]; or oxides like $ZrO_2$ [24–28]. $MoSi_2$ and SiC are the most commonly used sintering activators for boride ceramics including $TiB_2$ [2,4,13,15–17,21]. The effect of silicide additives on the sinterability and properties of $TiB_2$ was, among others, studied by Raju, Murthy et al. [17,19]. Through the introduction of 2.5% $MoSi_2$ additive and the use of the hot-pressing technique, the authors of the discussed papers obtained dense composites at 1700 °C. According to the authors, $MoSi_2$ removes oxide impurities and then deforms the plasticity due to high temperature, filling the spaces between $TiB_2$ grains. In another paper, Murthy et al. [4] investigated the materials sintered by hot pressing with 0 to 25% $MoSi_2$ addition, leading to samples with density close to 98%, grain sizes of 2–5 μm, high hardness of 25–27 GPa and fracture toughness of 5.1 MPa·m$^{0.5}$. The optimum amount of $MoSi_2$ was 10%, which resulted in a fine-grained microstructure of the sinters and hardness of 27 GPa.

Although carbon is a well-known additive that activates the sintering of covalent ceramics, it has been only marginally investigated for borides [29–31]. Also, the combination of carbon additives with other additives has been rarely studied [32–35]. Among others, Khoeini et al. [34] studied the effect of SiC and carbon on the sinterability of $ZrB_2$. The authors found that the addition of SiC effectively activates the sintering, but its simultaneous use with a small amount of carbon (1–2%) led to a lower addition of SiC and density close to 100%.

In the present study, it was decided to extend the state of knowledge on $TiB_2$ sintering. Investigations related to the influence of a combination of $MoSi_2$ and carbon additives on the sinterability of $TiB_2$ were carried out. The resulting high-density sinters were then tested with regard to their mechanical properties.

## 2. Materials and Experimental Procedure

The samples were prepared from commercial powders: $TiB_2$, ABCR Company, Germany (GRADE F, cat.no AB 134577); $MoSi_2$, Morton Thiokol, USA (99%, cat.no 48108). As a carbon precursor, phenol-formaldehyde resin of the NOVOLAK type (Chemical Plants Organika, Organika-Sarzyna, Poland) was used. During sintering, it undergoes pyrolysis, leaving 50 wt.% of amorphous carbon.

A reference sample of pure $TiB_2$ and three series of samples with various additive additions were made. The first series featured the addition of 1%, 2%, 3% and 4% wt. carbon; the second with the addition of 2.5%, 5% and 10% wt. $MoSi_2$; and the third one containing a constant amount of 2% wt. of carbon and 2.5%, 5% or 10% wt. of $MoSi_2$. The sample denominations used in this work are summarised in Table 1.

**Table 1.** Denomination of the samples.

| The Initial Composition | Name |
|:---:|:---:|
| $TiB_2$ | TB_0 |
| $TiB_2$ + 1% C | TB_1C |
| $TiB_2$ + 2% C | TB_2C |
| $TiB_2$ + 3% C | TB_3C |
| $TiB_2$ + 4% C | TB_4C |
| $TiB_2$ + 2.5% $MoSi_2$ | TB_2.5MS |
| $TiB_2$ + 5% $MoSi_2$ | TB_5MS |
| $TiB_2$ + 10% $MoSi_2$ | TB_10MS |
| $TiB_2$ + 2% C + 2.5% $MoSi_2$ | TB_2C_2.5MS |
| $TiB_2$ + 2% C + 5% $MoSi_2$ | TB_2C_5MS |
| $TiB_2$ + 2% C + 10% $MoSi_2$ | TB_2C_10MS |

The powder mixture components were weighed and then homogenised in ethanol in a ball mill for 12 h, using SiC spherical grinding media. Then, the alcohol was evaporated, and the powder mixtures were granulated by passing through a nylon 6 sieve. Cylindrical samples with a diameter of 12 mm and height 3–4 mm were formed through uniaxial double-ended pressing and then subjected to the dilatometric analysis.

The dilatometric sintering analysis was performed in a high-temperature graphite dilatometer of the authors' own construction. Sintering in the dilatometer was carried out in an argon flow with a heating rate 10 °C/min. The end of sintering happened when a characteristic *plateau* appeared on a linear dimensional change as a function of temperature. The dilatometric analysis enabled the final temperature of hot pressing to be determined.

The granulated powders were hot pressed using graphite dies under argon flow using Thermal Technology Inc. press model HP916-G-G. The reference sample and the samples with carbon addition were sintered at 25 MPa at 2100 °C, and the samples with $MoSi_2$ and $MoSi_2$ and carbon addition were sintered at 25 MPa at 1800 °C. All samples were kept at the final temperature for one hour. The heating rate in each case was 10 °C/min.

Apparent density of the sintered samples was measured using the Archimedes method. The relative density was calculated using 4.52 g/cm$^3$ as the theoretical density of $TiB_2$. The surface of the samples was ground and polished using LaboPol (Struers, Champigny, France) polishing machine. Their microstructure was analysed using Scios2 DualBeam (ThermoFisher, Waltham, MA, USA) SEM microscope along with the EDS chemical analysis. The observations were made using the Circular Backscatter Electrons (CBS) detector.

In order to determine the phase composition, XRD analysis was performed with the X'Pert Pro apparatus (PANanlitycal, Malvern, UK). The quantitative phase composition of the sinters was determined using the Rietveld method. Hardness measurements were carried out by the Vickers method, using a FV-810 (Future-Tech, Tokyo, Japan) hardness tester. A standard load of 1 kg and an indenter pressing time of 10 s were used. Fracture toughness ($K_{Ic}$) was determined using the indentation method at 3 kg load. The Niihara formula (Equation (1)) was used to calculate the critical stress intensity factor ($K_{Ic}$).

$$K_{Ic} = 0.018 \cdot \frac{HV^{0.6} \cdot E^{0.4} \cdot 0.5d}{l^{0.5}} \tag{1}$$

where:

E—Young's modulus, MPa
HV—Vickers hardness, MPa
d—the indentation diagonal, m
l—mean radial cracks length, m

Young's modulus measurements were carried out using the ultrasonic method by measuring the velocity of transverse ($C_T$) and longitudinal ($C_L$) waves passing through the specimen with EPOCH 3 (Panametrics) ultrasonic defectoscope. The defectoscope, equipped with broadband ultrasonic transducers for longitudinal and transverse waves, was used to measure the transit times of the ultrasonic waves through the specimen. The transducers, characterised by short pulse durations, enable measurements on the specimens with small thicknesses to be carried out, eliminating overlapping of the successive pulses. Young's modulus was calculated from the velocities of ultrasonic wave propagation in the sample and density of the material (Equation (2)).

$$E = \rho \cdot C_T^2 \frac{3C_L^2 - 4C_T^2}{C_L^2 - C_T^2} \tag{2}$$

where:

E—Young's modulus, GPa
$C_L$—velocity of the longitudinal ultrasonic wave, km/s
$C_T$—velocity of the transverse ultrasonic wave, km/s
$\rho$—apparent density of the material, g/cm$^3$

## 3. Results and Discussion

### 3.1. Dilatometric Analysis

Figure 1 shows the dilatometric sintering (sample shrinkage vs. temperature) curves, obtained for a series of samples with carbon addition and the reference sample. Sintering in the dilatometer was carried out up to 2150 °C, and despite such a high sintering temperature, sintering curves of the reference sample and the sample with 1 wt.% carbon addition do not show the *plateau* characteristic for the end of sintering.

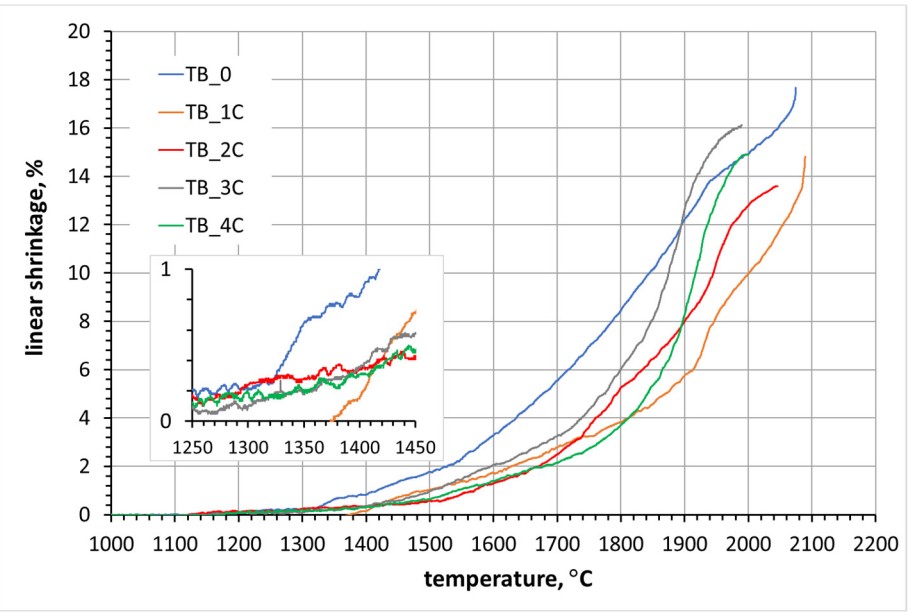

**Figure 1.** Dilatometric sintering curves of the reference sample and samples with carbon addition.

Flattening of the sintering curve can be observed for the samples with a carbon addition higher than 1 wt.%. Sintering of the samples with carbon additions between 2 and 4 wt.% ends at ca. 2050 °C.

Figure 2 presents the sintering curves of the samples with $MoSi_2$ and $MoSi_2$ and carbon addition.

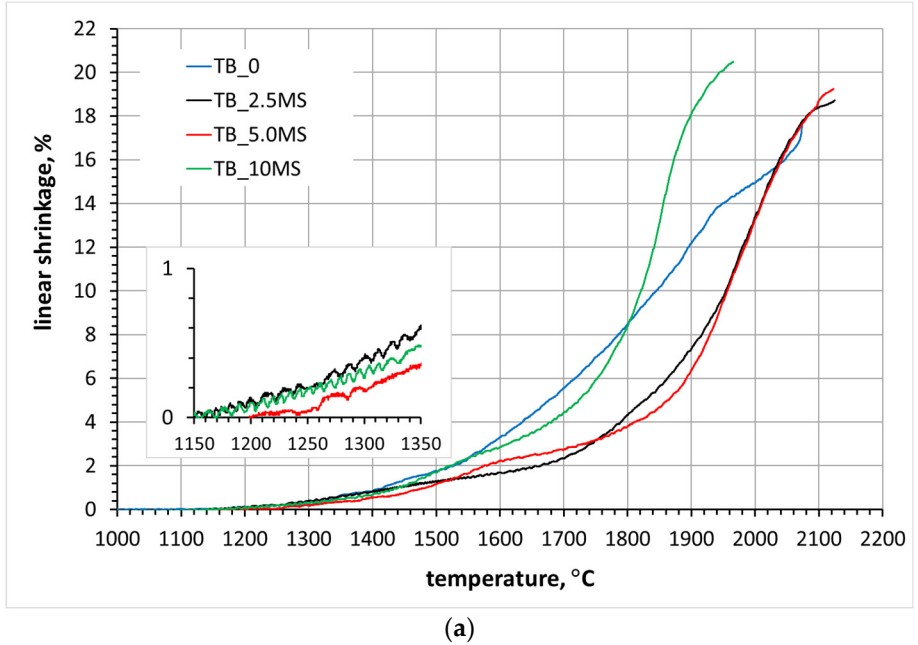

(**a**)

**Figure 2.** *Cont*.

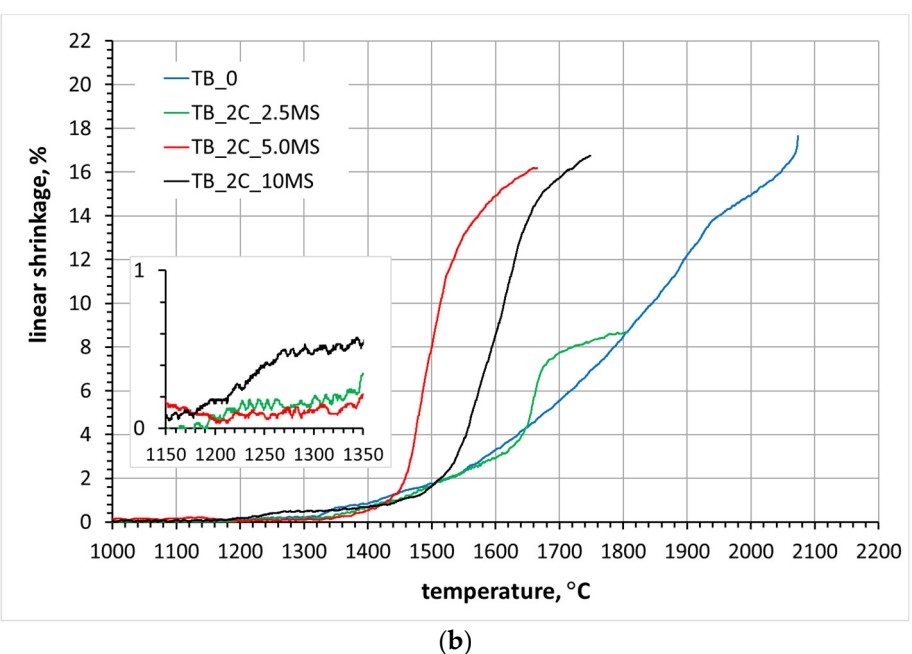

**(b)**

**Figure 2.** Dilatometric sintering curves of samples with (**a**) MoSi$_2$, (**b**) carbon and MoSi$_2$ additive.

The course of curves of the samples with MoSi$_2$ indicate that this addition effectively activates TiB$_2$ sintering (Figure 2a). When the MoSi$_2$ additive is 2.5 and 5 wt.%, the end of sintering occurs near 2100 °C, while a significant reduction in sintering temperature occurs when the additive is 10 wt.% because the temperature is close to 1950 °C. The best sintering results are given by the combination of carbon and MoSi$_2$ additives. For the samples containing 2 wt.% of carbon and 5 or 10 wt.% of MoSi$_2$, the characteristic *plateau*, indicating the end of sintering, occurs around 1700–1750 °C. The sample containing 2 wt.% carbon and 2.5 wt.% MoSi$_2$ addition shows the lowest linear shrinkage 8%, and its sintering ends near 1800 °C (Figure 2b). The beginning of sintering of the reference sample and the samples with carbon addition is in a range of 1300–1400 °C, while for the samples with MoSi$_2$ and carbon and MoSi$_2$ addition, it occurs in a temperature range of 1200–1350 °C. Based on the results of the dilatometric measurements, the temperature of hot pressing for the different samples was established.

### 3.2. Sintering of TiB$_2$ with Various Amounts of Carbon

The samples with carbon addition and the reference sample were sintered by hot pressing (HP) at 2100 °C. The relative density of the sinters is shown in Figure 3.

The reference sample shows the lowest relative density of 88%, and the introduction of 1 wt.% of carbon addition significantly increases the density (Figure 3). The highest density, around 98%, is achieved by the sinter with 2 wt.% carbon addition, while the density of the sinters with 3 and 4 wt.% carbon additions slightly decreases.

Figure 4 shows SEM microphotographs of the reference sample, the microstructure of which correlates with the measured density. The sample microstructure is inhomogeneous, and significant porosity is visible (black areas).

Figure 5 shows microstructures of the samples with carbon addition between 1% and 4%. The microstructures are homogeneous and characteristic for dense sinters. Black areas visible in the microstructure may be pores but also carbon or carbide inclusions, as confirmed by the EDS analysis (Figure 6).

In view of the XRD phase composition analysis, all the obtained polycrystals consist of 100% of TiB$_2$. Carbon and carbides may not be detected, as their amount in the composites may lay below the detection threshold of the XRD method.

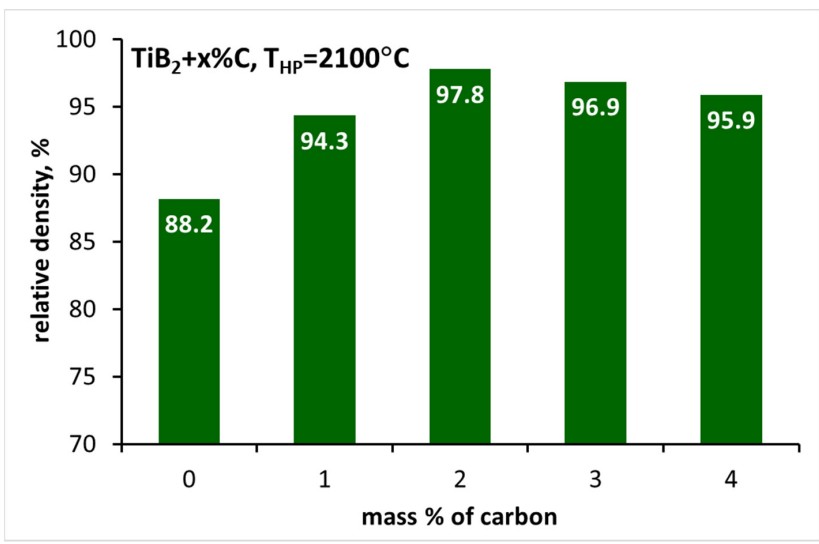

**Figure 3.** Relative density of the sintered samples containing various amounts of carbon.

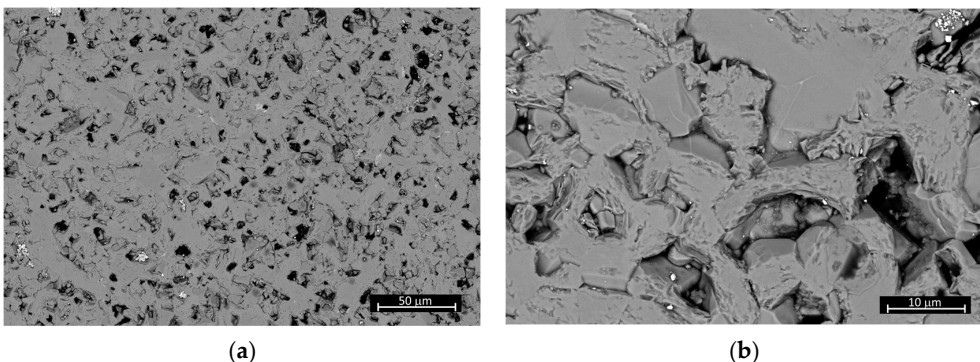

(**a**)                                    (**b**)

**Figure 4.** SEM images of the reference TiB2 sample (TB_0) at low (**a**) and high (**b**) magnification.

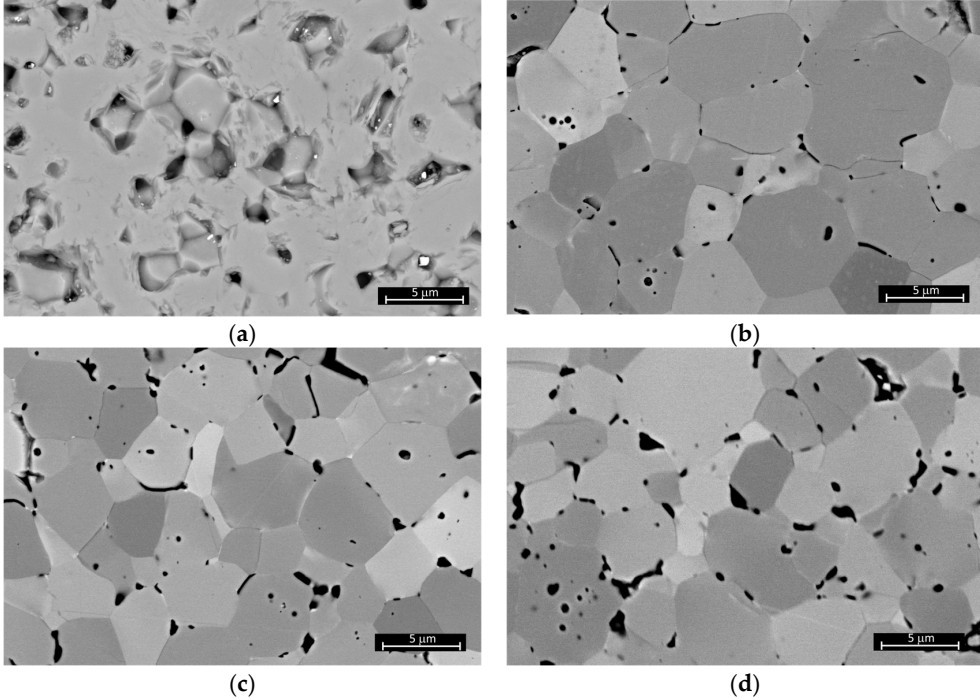

(**a**)                                    (**b**)

(**c**)                                    (**d**)

**Figure 5.** SEM images of $TiB_2$ samples with carbon addition: (**a**) 1% C; (**b**) 2% C; (**c**) 3% C; (**d**) 4% C.

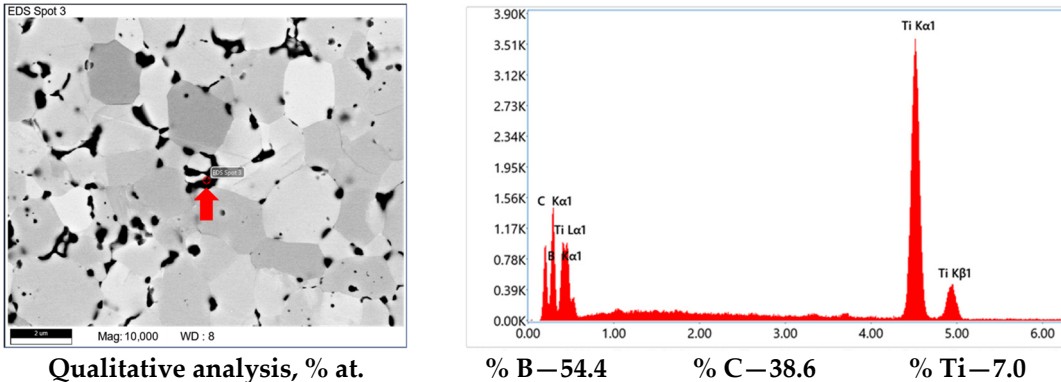

**Figure 6.** Chemical analysis of the black areas in the $TiB_2$ sample with 4 wt.% carbon addition.

### 3.3. Sintering of $TiB_2$ with Various Amounts of $MoSi_2$

$TiB_2$ samples with $MoSi_2$ additive were hot pressed at 25 MPa at 1800 °C. Figure 7 shows the dependence of relative density of the sinters on the amount of $MoSi_2$ additive. The lowest density, slightly lower than that of the reference sample, is shown by the composite with 2.5 wt.% $MoSi_2$ addition. A significant increase in density is observed for the samples with 5 and 10 wt.% addition of $MoSi_2$. Their relative density is 100%, which clearly indicates good sintering activation of $TiB_2$ by $MoSi_2$ (Figure 7).

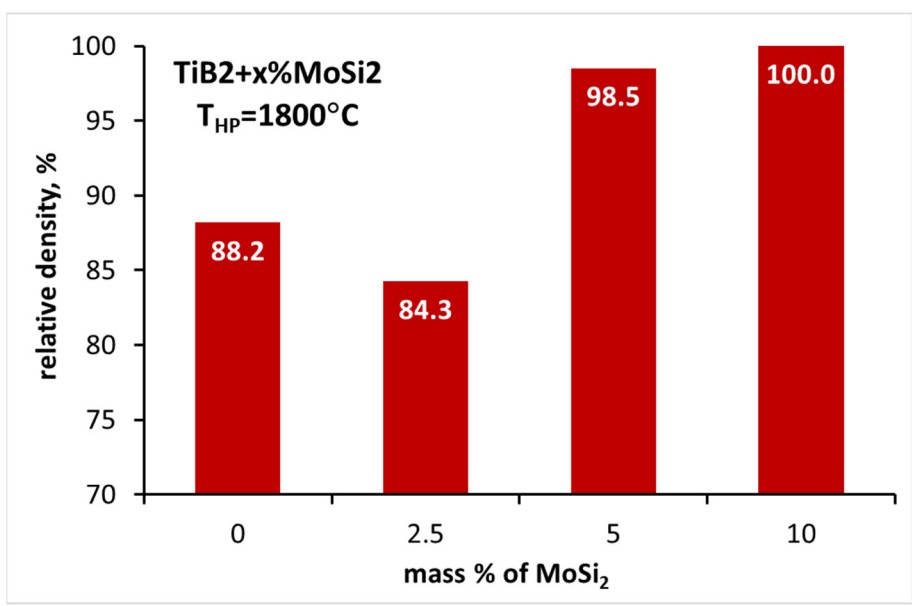

**Figure 7.** Relative density of the sintered samples containing various amounts of $MoSi_2$.

Figure 8 shows SEM images of the $TiB_2$ composites containing $MoSi_2$. Their microstructures correlate well with the relative density. The highest porosity is found in the sample with the lowest amount, i.e., 2.5 wt.% of $MoSi_2$. The micrographs of the sample clearly show the presence of pores (the darkest areas) (Figure 8a). The number of black areas indicating the presence of pores in the samples with 5 and 10 wt.% $MoSi_2$ addition is relatively low (Figure 8b,c). It can be said that pores are absent in the sample with 5 wt.% $MoSi_2$ addition (Figure 8b).

Based on the XRD phase composition analysis (Table 2), it can be concluded that $TiB_2$ dominates in all samples. In the samples containing 2.5 wt.% and 5 wt.% $MoSi_2$, negligible amounts of MoC are also identified. In contrast, $MoSi_2$ and MoC can be identified in the sample containing 10 wt.% $MoSi_2$. The presence of molybdenum carbides is the result of a reaction between $MoSi_2$, oxide impurities and carbon from the graphite foil and the graphite die (HP).

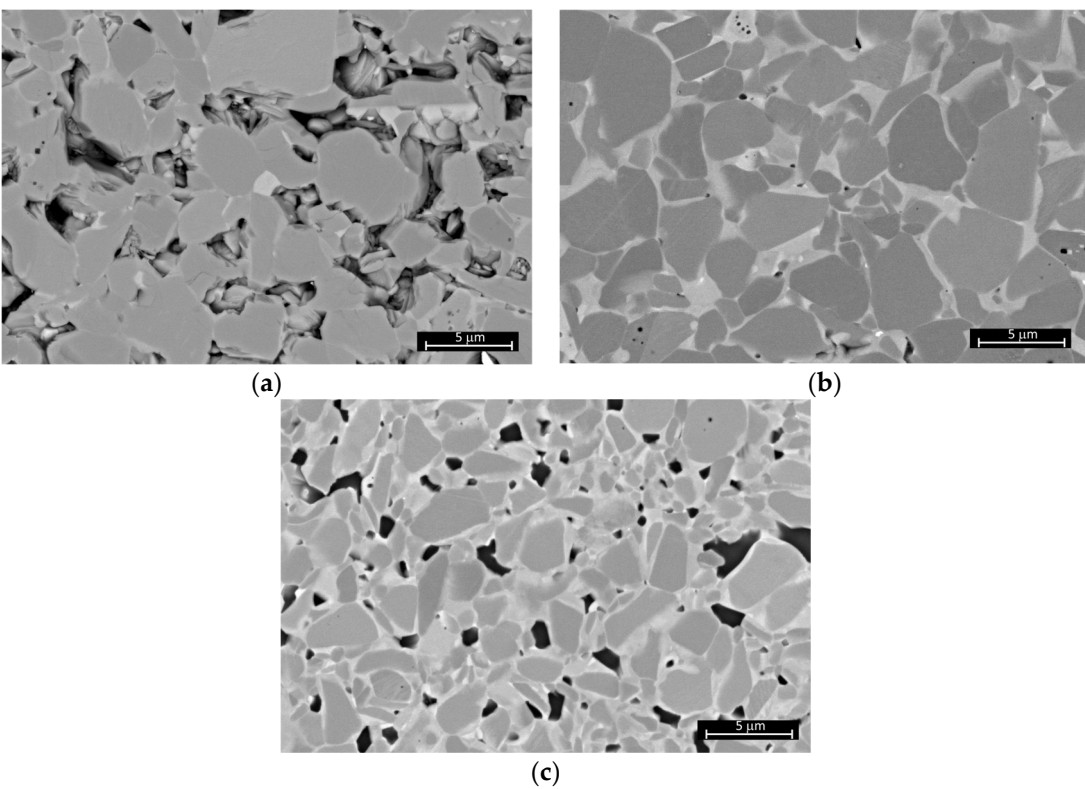

**Figure 8.** SEM microphotographs of TiB$_2$ samples sintered with various amounts of MoSi$_2$ addition: 2.5 wt.% (**a**); 5.0 wt.% (**b**) and 10 wt.% (**c**).

**Table 2.** Quantitative phase composition of TiB$_2$ + MoSi$_2$ composites.

| ICSInitial Phase Composition, wt.% | Phase Composition of the HP Sinters, wt.% |
|---|---|
| 97.5% TiB$_2$, 2.5% MoSi$_2$ | 68.8% TiB$_2$ 1, 29.7% TiB$_2$ 2, 1.5% MoC |
| 95% TiB$_2$, 5.0% MoSi$_2$ | 69.8% TiB$_2$ 1, 28.7% TiB$_2$ 2, 1.5% MoC |
| 90% TiB$_2$, 10% MoSi$_2$ | 83.0% TiB$_2$ 1, 9.4% TiB$_2$ 2, 1.0% MoC, 6.6% MoSi$_2$ |

Furthermore, in the light of the XRD analysis, titanium borides with the same structure but different lattice parameters are present in the composites (Table 3, TiB$_2$ **1** and TiB$_2$ **2**).

**Table 3.** Lattice parameters of titanium boride phases identified in TiB$_2$ + MoSi$_2$ composites.

| Lattice Parameter, Å | Theoretical Unit Cell Parameters of TiB$_2$, [36] | TiB$_2$ + 2.5% MoSi$_2$ | | TiB$_2$ + 5.0% MoSi$_2$ | | TiB$_2$ + 10% MoSi$_2$ | |
|---|---|---|---|---|---|---|---|
| | | TiB$_2$ 1 | TiB$_2$ 2 | TiB$_2$ 1 | TiB$_2$ 2 | TiB$_2$ 1 | TiB$_2$ 2 |
| a | 3.028 | 3.030 | 3.028 | 3.030 | 3.030 | 3.029 | 3.029 |
| b | 3.028 | 3.030 | 3.028 | 3.030 | 3.030 | 3.029 | 3.029 |
| c | *3.228* | *3.230* | *3.230* | *3.230* | *3.231* | *3.231* | *3.230* |

TiB$_2$ 1 and TiB$_2$ 2—TiB$_2$ with the same structure but different lattice parameters.

The differences in hues of grey visible in the SEM images indicate differences in the chemical composition of individual areas of the samples. Figure 9 shows the result of a local chemical composition analysis of the TiB$_2$ composite containing 10 wt.% MoSi$_2$, and the results of the chemical composition analysis are consistent with the results of the phase composition analyses of the sample (Table 2).

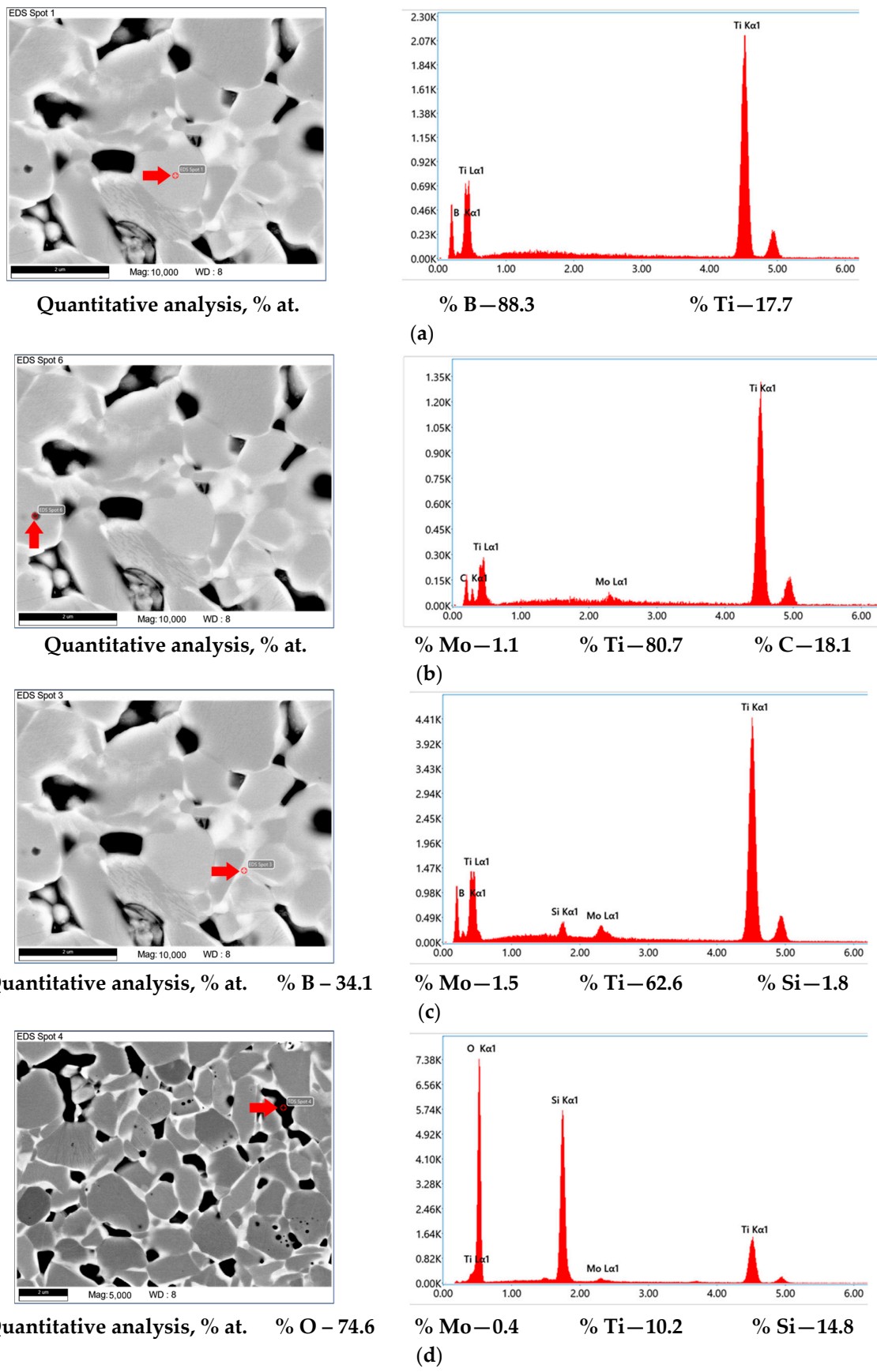

**Figure 9.** Results of the point chemical analysis EDS of TiB$_2$ samples with 10 wt.% MoSi$_2$ addition (**a–d**) (chemical analysis was carried out at the points marked by red arrows on the SEM images).

In the sinters, grains cores rich in titanium and boron can be identified, which are most likely titanium diboride (Figure 9a). There are also areas in which, in addition to titanium, molybdenum and carbon (Figure 9b) as well as molybdenum and silicon (Figure 9c) can be found. An increased amount of molybdenum is identified in light-grey areas around $TiB_2$ grains (Figure 9c). There are also the darkest areas, which, in many cases, are pores but not always, as Figure 9d shows. In many such areas, significant amounts of oxygen and silicon can be identified. The reaction between $MoSi_2$ and the oxides which passivate the boride grains can result in the formation of an amorphous phase from the Si-O-B system [37–39].

### 3.4. Sintering of $TiB_2$ with 2 wt.% Carbon Addition and Various Amounts of $MoSi_2$

$TiB_2$ samples, containing 2 wt.% carbon and different $MoSi_2$ additions, were hot pressed at 1800 °C. The relative densities of the resulting polycrystals are shown in Figure 10.

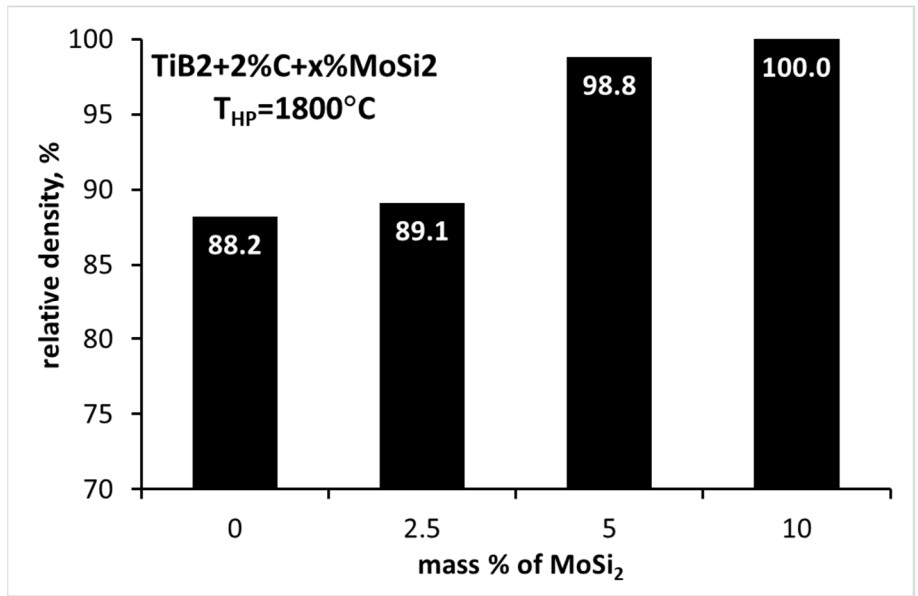

**Figure 10.** Relative density of sintered samples with 2 wt.% carbon and various amounts of $MoSi_2$.

The lowest density ca. 90% was exhibited by the sample with the lowest $MoSi_2$ content. An increase in $MoSi_2$ addition to 5 wt.% led to composites with densities higher than 98%. The relative density of 100% was achieved by the composite with the highest $MoSi_2$ addition, i.e., 10 wt.% (Figure 10).

The combination of carbon and $MoSi_2$ additions results in significantly increased density of the composite with 2.5 wt.% $MoSi_2$ addition, as compared to the analogous composite without carbon addition (Figure 7).

Figure 11 shows SEM microstructures of the $TiB_2$ samples with a simultaneous addition of carbon and $MoSi_2$. In the sample with 2.5 wt.% $MoSi_2$ addition, significant porosity is visible (darkest areas). A homogeneous, dense microstructure is presented by the samples with 5 and 10 wt.% $MoSi_2$ additions (Figure 11b,c). Again, the microstructure of $TiB_2$ composites with carbon and different amounts of $MoSi_2$ addition is similar to the one typical for cermets [40], i.e., grains consisting of cores and characteristic shells (Figure 11).

According to the results of XRD phase composition analysis, the composites after sintering are dominated by titanium boride with different elemental cell sizes (Tables 4 and 5). The presence of two hexagonal $TiB_2$ phases (here named $TiB_2$ 1 and $TiB_2$ 2) with different lattice parameters may indicate substitutions within the boride cell by the additive-derived elements, i.e., Mo, Si and C. Furthermore, $MoSi_2$ is not present after sintering. It is likely that during sintering, chemical reactions occur between $TiB_2$, the oxide impurities,

i.e., $TiO_2$ and $B_2O_3$, and the additives, i.e., $MoSi_2$ and carbon. These reactions resulted in the formation of silicon carbide and complex carbide $(Ti, Mo)C_2$ (Table 4).

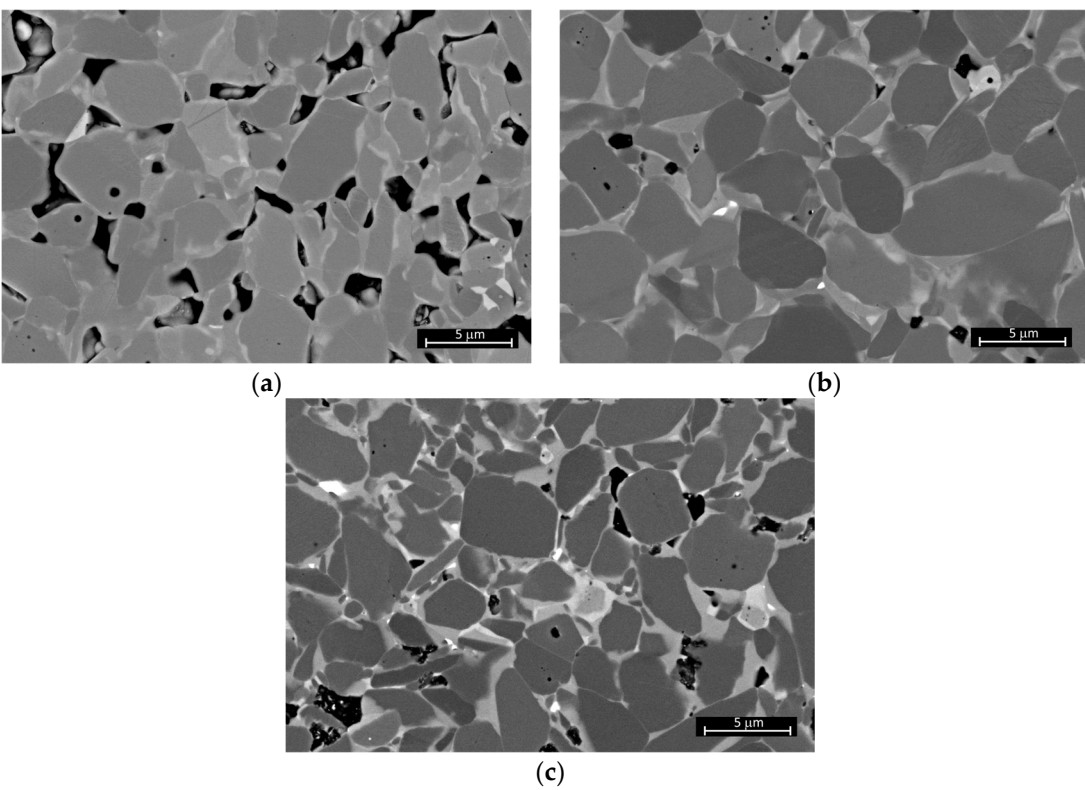

**Figure 11.** SEM images of $TiB_2$ samples with 2% carbon and different amounts of $MoSi_2$ addition: 2.5% (**a**), 5.0% (**b**) and 10% (**c**).

**Table 4.** Quantitative phase composition of $TiB_2$ + 2%C + x%$MoSi_2$ composites.

| Initial Phase Composition, wt.% | Phase Composition of the HP Sinters, wt.% |
|---|---|
| 95.5% $TiB_2$, 2.0% C, 2.5% $MoSi_2$ | 96.8% $TiB_2$ 1, 0.1% $TiB_2$ 2, 1.7% TiC, 1.4% SiC |
| 93% $TiB_2$, 2.0% C, 5.0% $MoSi_2$ | 66.3% $TiB_2$ 1, 28.9% $TiB_2$ 2, 1.8% SiC, 3.0% $(Ti,Mo)C_2$ |
| 88% $TiB_2$, 2.0% C, 10% $MoSi_2$ | 76.1% $TiB_2$ 1, 18.6% $TiB_2$ 2, 3.1% SiC, 2.2% $(Ti,Mo)C_2$ |

**Table 5.** Lattice parameters of titanium boride phases identified in $TiB_2$ + 2% C + x% $MoSi_2$ composites.

| Lattice Parameter, Å | Theoretical Unit Cell Parameters of $TiB_2$, [36] | $TiB_2$ + 2%C +2.5% $MoSi_2$ | | $TiB_2$ + 2%C +5.0% $MoSi_2$ | | $TiB_2$ + 2%C +10% $MoSi_2$ | |
|---|---|---|---|---|---|---|---|
| | | $TiB_2$ 1 | $TiB_2$ 2 | $TiB_2$ 1 | $TiB_2$ 2 | $TiB_2$ 1 | $TiB_2$ 2 |
| a | 3.028 | 3.027 | 3.034 | 3.029 | 3.027 | 3.029 | 3.027 |
| b | 3.028 | 3.027 | 3.034 | 3.029 | 3.027 | 3.029 | 3.027 |
| c | 3.228 | 3.233 | 3.220 | 3.230 | 3.231 | 3.230 | 3.232 |

$TiB_2$ 1 and $TiB_2$ 2—$TiB_2$ with the same structure but different lattice parameters.

The EDS chemical analysis of the selected sample (TB_2C_10MS) is shown in Figure 12. The darkest areas could be pores but also oxide or carbide grains since the areas are rich in oxygen, carbon, silicon and titanium as well as molybdenum. The dark-grey areas (cores) are most likely $TiB_2$ grains, while the light-coloured grain shells and grain boundary areas are rich in molybdenum, titanium, boron and, in some places, also carbon.

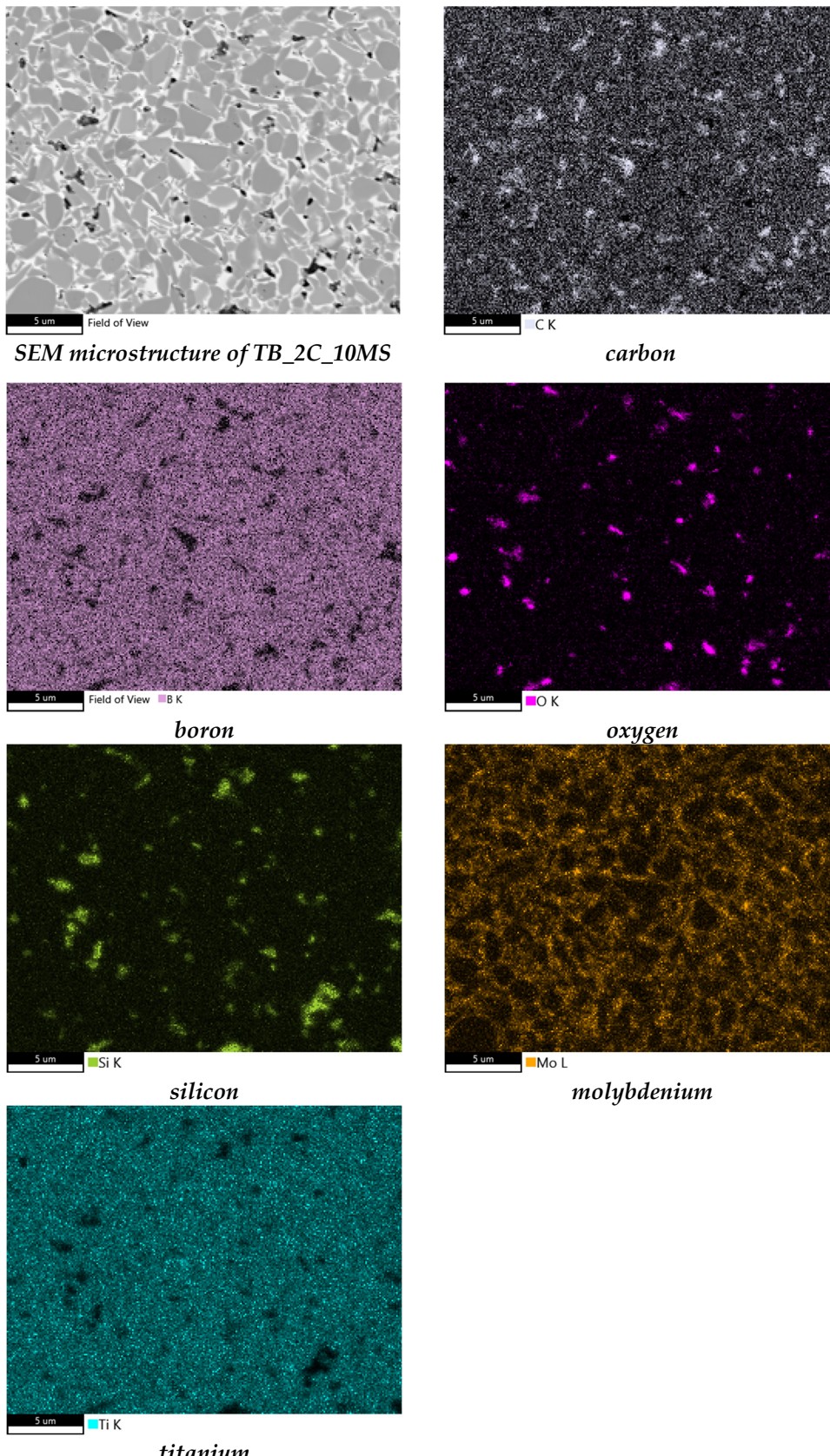

*SEM microstructure of TB_2C_10MS*

*carbon*

*boron*

*oxygen*

*silicon*

*molybdenium*

*titanium*

**Figure 12.** EDS element distribution maps of the TB_2C_10MS composite.

### 3.5. Mechanical Properties of the TiB$_2$-MoSi$_2$-C Ceramics

The composites were tested for Vickers hardness, critical stress intensity factor (K$_{Ic}$), which is a measure of fracture toughness, and Young's modulus. Due to the highly subjective measurement of the critical stress intensity factor using the indentation method, this was carried out on the composites with a relative density higher than 95%. The obtained results are summarised in Table 6.

**Table 6.** Relative density and selected mechanical properties of the materials.

| Sample | Sintering Temperature (HP), °C | Relative Density, % (*) | Vickers Hardness, GPa | Fracture Toughness, MPa·m$^{0.5}$ | Young's Modulus, GPa |
|---|---|---|---|---|---|
| TB_0 | 2150 | 88.2 ± 0.3 | 19.09 ± 6.30 | - | - |
| TB_1C | 2150 | 94.3 ± 0.1 | 26.31 ± 5.86 | - | 526 ± 12 |
| TB_2C | 2150 | 97.8 ± 0.1 | 25.31 ± 0.77 | 5.16 ± 0.28 | 536 ± 9 |
| TB_3C | 2150 | 96.9 ± 0.4 | 23.34 ± 2.17 | 5.26 ± 0.47 | 496 ± 16 |
| TB_4C | 2150 | 95.9 ± 0.3 | 25.68 ± 5.29 | 5.52 ± 0.20 | 542 ± 10 |
| TB_2.5MS | 1800 | 84.3 ± 0.6 | 16.97 ± 2.86 | - | - |
| TB_5MS | 1800 | 98.5 ± 0.2 | 26.21 ± 2.25 | 6.25 ± 0.51 | 536 ± 11 |
| TB_10MS | 1800 | 100.0 ± 0.4 | 26.78 ± 3.37 | 4.86 ± 0.19 | 504 ± 24 |
| TB_2C_2.5MS | 1800 | 89.1 ± 0.8 | 17.19 ± 1.87 | - | 440 ± 14 |
| TB_2C_5MS | 1800 | 98.8 ± 0.4 | 24.88 ± 2.03 | 4.79 ± 0.52 | 543 ± 6 |
| TB_2C_10MS | 1800 | 100.0 ± 0.2 | 24.41 ± 1.90 | 4.17 ± 0.31 | 533 ± 12 |

(*) the theoretical density of TiB$_2$ was 4.52 g/cm$^3$.

The hardness of the samples with the lowest densities, i.e., the reference sample (TB_0), with 2.5% MoSi$_2$ addition (TB_2.5MS) and with 2% carbon and 2.5% MoSi$_2$ addition, is not higher than 20 GPa (Table 6). The introduction of 1% carbon results in a noticeable increase in hardness to 26 GPa. The hardness of composites with 1 to 4% carbon addition is similar and ranges from 23 to 26 GPa. For the composites with MoSi$_2$ addition, the lowest hardness is shown by TB_2.5MS and TB_2C_2.5MS samples. For other composites with MoSi$_2$ addition, the hardness ranges from 24 to 26 GPa.

Based on the measured values of the critical stress intensity factor (Table 6), it can be concluded that the composites exhibit high fracture toughness. The lowest K$_{Ic}$ values, larger than 4 MPa·m$^{0.5}$, are shown by the composites with both additives, i.e., carbon and MoSi$_2$ additives. The K$_{Ic}$ values, close to 5 MPa·m$^{0.5}$, are exhibited by the composites with carbon additions of 2 to 4%. In contrast, K$_{Ic}$ values between 4.86 and 6.25 MPa·m$^{0.5}$ are shown by the composites with the MoSi$_2$ additive.

The Young's modulus values of all composites are high, ranging from 496 GPa for the TB_3C composite to 543 GPa for the TB_2C_5MS composite.

## 4. Results and Discussion

On the basis of the results of these investigations, a favourable effect of all used additives, i.e., carbon, MoSi$_2$ and MoSi$_2$+C, on the sinterability of TiB$_2$ titanium diboride was found. Firstly, a dilatometric sintering analysis was carried out, which showed the validity of the abovementioned sintering activators (Figures 1 and 2). Furthermore, it is clear from the dilatometric measurements that the use of 10% MoSi$_2$ addition and the combination of both additives significantly reduces the sintering temperature of TiB$_2$ (Figure 2). The sintering temperature of polycrystals with carbon and MoSi$_2$ additives does not exceed 1800 °C. For the polycrystals with MoSi$_2$, only a 10% addition of MoSi$_2$ reduces the sintering temperature to 1900 °C (Figure 2).

According to the dilatometric sintering analysis, the hot-pressing temperatures of all composites were determined. The composites with carbon as well as the reference sample were hot pressed at 2100 °C, while the composites with $MoSi_2$ and the composites with carbon and $MoSi_2$ were hot pressed at 1800 °C. In many cases, polycrystals with a relative density of 100% were obtained. In the case of carbon addition, the highest densities were obtained when the carbon addition was between 2 and 4 wt.%. Dense polycrystals were produced by adding 5 or 10 wt.% $MoSi_2$ as well as 2 wt.% carbon together with 5 or 10 wt.% $MoSi_2$ (Figures 3, 7 and 10).

Most of the obtained polycrystals (sinters) were non-porous, as can be seen from the microstructure analysis (Figures 5, 8 and 11), while it cannot be deduced from the apparent density measurements and relative density calculations (Table 6). One of the main reasons for the precise calculation of the theoretical density is the numerous reactions that take place during sintering between $TiB_2$, $MoSi_2$, carbon and oxide impurities, resulting in the formation of new phases, as confirmed by the results of the phase composition analysis (Tables 2 and 4). Therefore, taking the density calculated from the initial compositions as the theoretical density of the material in question appears to be incorrect. The calculation of the theoretical density from the phase composition analysis, due to the presence of solid solutions and amorphous phases, is also erroneous. For the sake of comparison, a single value of the theoretical density was used, and the theoretical density of $TiB_2$, i.e., 4.52 $g/cm^3$, was taken as the theoretical density. High density of the sinters was indirectly evidenced by water absorption measurements (Table 7). Water absorption of many polycrystals reaches only a few hundredths of one percent and is indicative of the lack of open porosity in these materials.

**Table 7.** Water absorbability of the materials.

| Sample | Water Asorbability, % |
|---|---|
| TB_0 | $0.03 \pm 0.01$ |
| TB_1C | $0.03 \pm 0.01$ |
| TB_2C | $0.01 \pm 0.02$ |
| TB_3C | $0.15 \pm 0.04$ |
| TB_4C | $0.08 \pm 0.03$ |
| TB_2.5MS | $3.69 \pm 0.50$ |
| TB_5MS | $0.05 \pm 0.02$ |
| TB_10MS | $0.04 \pm 0.01$ |
| TB_2C_2.5MS | $1.29 \pm 0.20$ |
| TB_2C_5MS | $0.08 \pm 0.02$ |
| TB_2C_10MS | $0.06 \pm 0.01$ |

The addition of carbon enables single-phase polycrystals to be obtained, as evidenced by the phase composition analysis, according to which only $TiB_2$ is present in such sinters. Also, the microstructures shown in Figure 5 are characteristic of single-phase and dense sinters when carbon addition is in a range of 2 to 4 wt.%. Phase composition analyses demonstrate the effectiveness of using carbon as an oxide impurity reducer. According to the literature [41], carbon can react both with $B_2O_3$ and $SiO_2$ at the temperatures close to 1000 °C in low vacuum (20 Pa). The local EDS analysis of the chemical composition, carried out during SEM observations, showed the trace presence of fine particles, whose chemical composition suggests that they may be the particles of carbon or boron and titanium carbides (Figure 6).

Carbon in the sintering of covalent ceramics acts as a reducer of oxide impurities [29–35]. When it was intentionally added, the reaction between oxide impurities, i.e., $SiO_2$ or $TiO_2$, resulted in the formation of SiC and TiC and complex titanium-molybdenum carbide,

respectively (Table 4). When carbon addition was not intentionally added, the favoured reaction was the formation of MoC. In this case, carbon can be regarded as an impurity from the system with which sintering was carried out, i.e., from the heating element, matrix or graphite film. In addition, in composites with only $MoSi_2$, oxide impurities are present in the form of an amorphous Si-O-B-Ti-Mo phase, which is not identified by XRD analysis but visible in SEM images (Figures 9d and 12).

The phase composition of the polycrystals sintered with $MoSi_2$ (Table 2) and with carbon and $MoSi_2$ additives shows that such composites (Table 4) were obtained, in which two $TiB_2$ phases with different lattice parameters dominated (Tables 3 and 5). In addition, molybdenum carbide was identified in all composites sintered with $MoSi_2$ only, and only the sample with the highest $MoSi_2$ addition showed the presence of this additive. In contrast, for the composites with both additives, the two titanium boride phases with different lattice parameters were dominant (Table 5, Supplementary Materials). Silicon and titanium carbides were also identified in these composites, and a complex carbide with the formula $(Mo,Ti)C_2$ was found in the samples with 5 and 10% wt. $MoSi_2$ additions. Substitutions of titanium ($a_r$ = 140 pm) by Mo ($a_r$ = 145 pm), Si ($a_r$ = 145 pm) cations as well as boron ($a_r$ = 75 pm) by carbon ($a_r$ = 70 pm) can led to the presence of hexagonal titanium boride phases with different lattice parameters or, in other words, to the presence of solid solutions [42]. The presence of $(Mo,Ti,Si)B_2$ solid solutions can be evidenced by a microstructure similar to that of *core–shell* cermets [40]. This type of microstructure is often found in composites based on metal borides in the fourth group of the periodic table of chemical elements [21–23,43–46]. SEM photographs of TB_MS and TB_C_MS composites (Figures 8 and 11) show microstructures similar to those typical for cermets. The microstructure of cermets typically consists of particles with "hard cores" and solid solution layers on their surface ("shells"). In typical cermets, such *core–shell* particles are joined together by bonding metals, e.g., cobalt [22,23,44,46]. With regard to the $TiB_2$-$MoSi_2$-C composites studied, it can be assumed that the cores are formed by one of the titanium boride phases with the shell by the titanium boride solid solution with substitutions (Figure 13).

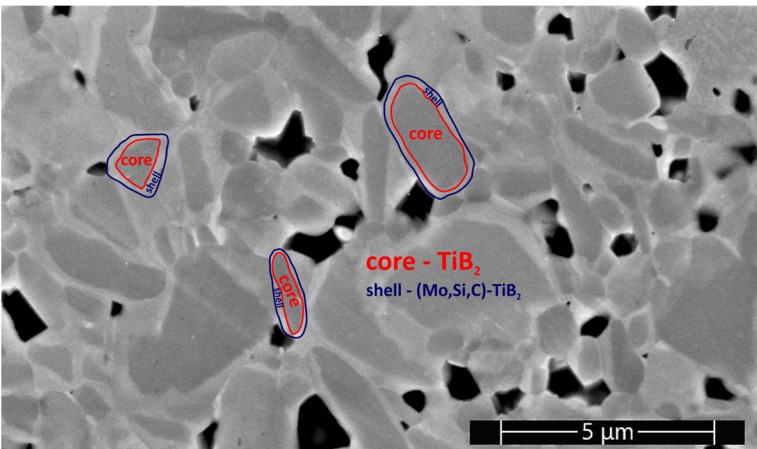

**Figure 13.** SEM microstructure of the composite with 10 wt.% $MoSi_2$ addition characteristics of cermets.

The EDS chemical element distribution maps, made during the SEM observations, showed further that, in both groups of composites with the $MoSi_2$ additive, there are areas enriched in oxygen and silicon (Figures 9 and 12). During sintering, a variety of reactions can occur in all composites, including the carbothermal reduction of oxides, passivating boride particles (Equations (3) and (4)), and the reactions between $MoSi_2$ and

oxide impurities (Equations (5)–(7)) [4,16,43,47,48], leading, among others, to the formation of silica, monoborides and carbides.

$$TiO_2 + 3C \rightarrow TiC + 2CO \uparrow \tag{3}$$

$$2B_2O_3 + 7C \rightarrow B_4C + 6CO \uparrow \tag{4}$$

$$2MoSi_2 + 2B_2O_3 + TiO_2 \rightarrow TiB_2 + 4SiO_2 + 2MoB \tag{5}$$

$$2MoSi_2 + 2B_2O_3 + 2.5C \rightarrow 2.5SiC + 1.5SiO_2 + 2MoB \tag{6}$$

$$5MoSi_2 + 7O_2 \rightarrow Mo_5Si_3 + 7SiO_2 \tag{7}$$

As shown by the composite microstructures in Figures 8 and 11, solid solutions can be formed at grain boundaries or, more commonly, at the surface of the boride grains. The literature reports that the formation of the solid solutions in question can be related to the presence of liquid phases with the compositions resulting from the initial chemical composition of the composites [1,17,21,44,47,49–51]. The most likely formation of liquid phases is from the Si–B–O system, in which the elements (Ti, Mo), forming the components of the composite, can dissolve. It should be added that in the case of the $MoSi_2$ additive alone, the passivating oxides do not reduce as readily as under the influence of carbon. During cooling, the epitaxial precipitation from the liquid phase and the formation of solid solutions can occur [4,17,21,44,51]. The occurrence of the solid solutions discussed, as well as silicide, carbide and $SiO_2$ phases, may be indirect evidence of the presence of liquid phases during the sintering of composites with $MoSi_2$ addition. Furthermore, oxygen-rich and silicon-rich particles were identified during the chemical composition analysis, even when $MoSi_2$ and carbon were used as additives (Figure 12).

According to the literature [37–39], the occurrence of liquid phases from the Si–B–O system is mainly possible in boride composites with silicide additives. These phases can effectively support sintering under pressure by facilitating, among other things, the movement of grains relative to each other [52]. In this case, if the oxides are not fully reduced, even when a small addition of carbon is introduced, it is possible for a reaction between $SiO_2$ and $B_2O_3$ to take place, resulting in the formation of a liquid phase from the Si–B–O system, in which elements present in the initial compounds forming the composites can dissolve.

Furthermore, the sintering temperature of composites with $MoSi_2$ is 1800 °C and does not exceed the melting point of the silicide ($T_m$ = 2050 °C) [50,53] but, as reported in the literature [52–55], at a temperature higher than 800 °C, silicides, including $MoSi_2$, deform plastically and can fill pores during sintering.

The relationship between the density of the composites and the values of the tested mechanical properties is observed. The composites with the highest density, regardless of the additive used, show high Vickers hardness of 23 to 26 Gpa. Also, in terms of fracture toughness, all the composites tested show a high value of the critical stress intensity factor $K_{Ic}$ (Table 6). The lowest values of $K_{Ic}$ from 4.18 to 4.79 Mpa·m$^{0.5}$ are exhibited by the composites with two additives, fracture toughness of composites with 2 to 4 wt.% carbon addition oscillates around 5 Mpa·m$^{0.5}$, while in the composites with $MoSi_2$, it ranges from 6.25 to 4.86 Mpa·m$^{0.5}$ for 5 wt.% $MoSi_2$ and 10 wt.% $MoSi_2$ addition, respectively. The typical phenomena leading to the increase in effective fracture energy are observed in composites, such as intergranular cracking (Figure 14a,c), crack deflection as well as crack defragmentation (Figure 14e,f). It is noteworthy that the grain boundaries (bonding phase) in the composites with 5 wt.% $MoSi_2$ addition are weaker than those in the composites with 10 wt.% $MoSi_2$ addition (compare Figure 14a with Figures 14b and 14c with Figure 14d). This results in a lower value of the critical stress intensity factor in the composites with 10 wt.% $MoSi_2$ addition (Table 6). The fracture then runs predominantly through the grains as well as along the $TiB_2$ grain boundaries (Figure 14b,d).

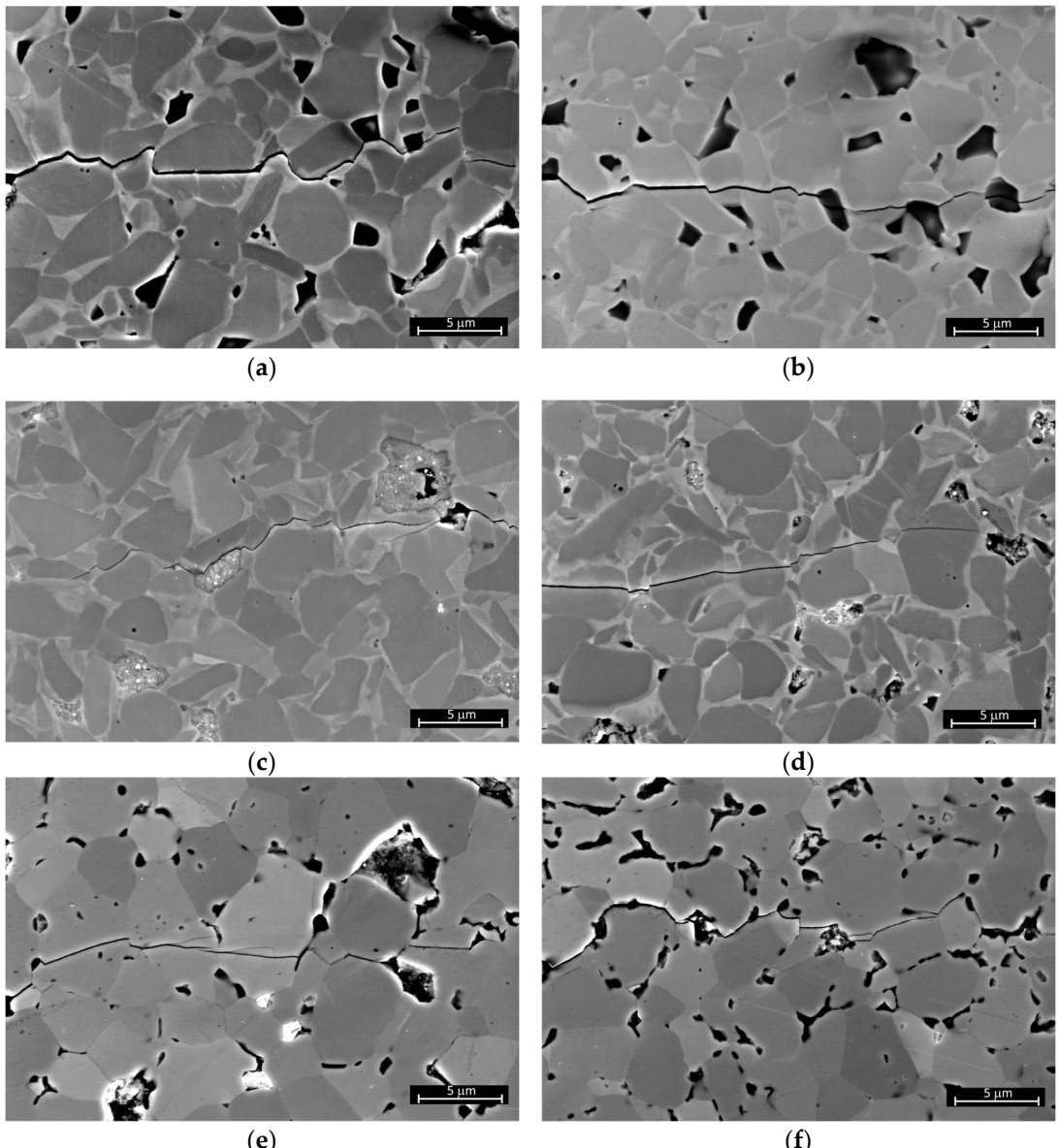

**Figure 14.** Course of cracks in composites: (**a**) TB2_5MS; (**b**) TB2_10MS; (**c**) TB_2C_5MS; (**d**) TB_2C_10MS; (**e**) TB_3C; and (**f**) TB_4C.

All high-density composites can be classified as low-deformability materials, as evidenced by the high values of their Young's modulus (Table 6).

The values of hardness, fracture toughness and Young's modulus shown for the investigated composites are similar to and often better than those reported in the literature [4,16,47,56].

## 5. Conclusions

1. The hot pressing of $TiB_2$ with $MoSi_2$ or carbon and $MoSi_2$ with carbon resulted in single-phase polycrystals and composites with density higher than 95%. Both additives used can be considered as $TiB_2$ sintering activators.

2. It is noteworthy that the use of $MoSi_2$ as a sintering activating additive significantly reduces the sintering temperature of titanium boride down to 1800 °C.

3. With the addition of carbon, it is possible to obtain single-phase polycrystals, in which only the $TiB_2$ phase is present. On the other hand, with the addition of $MoSi_2$ as well

as the combined addition of MoSi$_2$ and carbon, it is possible to obtain solid composites with a *core–shell* microstructure, characteristic for cermets.

4. Carbon is an effective reducer of oxide impurities during TiB$_2$ sintering.
5. Due to the presence of liquid phases from the Si–B–O–Mo system and plastic deformation of MoSi$_2$, it becomes possible to obtain dense composites based on TiB$_2$.
6. The produced materials have potential in high-temperature applications due to the high melting point of TiB$_2$ and very good properties, such as high hardness, high fracture toughness and low deformability. Nevertheless, further testing of their thermal and chemical properties is required.

**Supplementary Materials:** The following are available online at https://www.mdpi.com/article/10.3390/cryst14030212/s1, Figure S1: An example of a reflection indicating the presence of two titanium borides with different lattice parameters, Figure S2: Differences between the actual (red curve) and theoretical (blue curve) curves, indicating a significant error on the XRD analysis, Figure S3: High theoretical and real curve fit in XRD analysis, Figure S4. XRD pattern of TB_0 sample, Figure S5. XRD pattern of TB_4C sample, Figure S6. XRD pattern of TB_10MS composite and Figure S4. Figure S7, XRD pattern of TB_2C_10MS composite.

**Author Contributions:** Conceptualization, A.G., Ł.Z., K.K. and M.S.; methodology, M.S., Ł.Z., A.G., K.K., N.M. and S.K.; validation, Ł.Z., D.Z. and K.K.; formal analysis, A.G., K.K., N.M., S.K. and E.G.; investigation, M.S., D.Z., K.K., K.M., N.M., S.K. and E.G.; writing—original draft preparation, D.Z., M.S. and A.G.; writing—review and editing, Ł.Z., K.K. and A.G.; visualization, A.G., M.S. and D.Z.; supervision, A.G. and Ł.Z. All authors have read and agreed to the published version of the manuscript.

**Funding:** This research was funded in whole or in part by the National Science Centre, Poland, registration number 2022/06/X/ST5/01119 Miniatura 6 "The influence of silicide additives on the sintering temperature and mechanical properties of ultra-high temperature AlB$_2$-type borides-based ceramics" (18.160.107). For the purpose of Open Access, the author has applied a CC-BY public copyright license to any Author Accepted Manuscript (AAM) version arising from this submission.

**Data Availability Statement:** The raw data supporting the conclusions of this article will be made available by the authors on request.

**Acknowledgments:** This work was carried out under a subsidy from the Ministry of Education and Science to the AGH University of Krakow (project no 16.16.160.557). The SEM investigations were supported by the program Excellence Initiative—Research University for the AGH University of Krakow, Grant ID 1449 (PI: M.Ziabka).

**Conflicts of Interest:** The authors declare no conflicts of interest.

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
