# Peer review of "Processing, Microstructure and Mechanical Properties of TiB2-MoSi2-C Ceramics"

_crystals, doi:10.3390/cryst14030212_

Round 1

Reviewer 1 Report

Comments and Suggestions for Authors

Dear Authors, look please into the attached pdf file for a !sticky notes! with my remarks. Particularly, I suggest a simplification in the sample labeling system. Various subtle modification should also be made.

Comments on the Quality of English Language

The language needs only very subtle changes, as indicated in the pdf file.

Reviewer 2 Report

Comments and Suggestions for Authors

In the paper "I will leave it to you to respond, but I recommend removing the low magnification SEM image due to the large number of figures", M. Sajdak and co-authors investigated the effects of carbon and MoSi2 on the TiB2 composite ceramics. The detailed comments are as follows.

1.       Please describe the motivation in Introduction why the amorphous carbon was added in this system.

2.       The relative density was calculated from 4.52 g/cm3, but the theoretical value varies depending on the composition, especially in the case of MoSi2 addition.

3.       The XRD curves should be added because the quantitative phase composition and lattice parameters were calculated from them.

4.       The author must describe the meaning of the TiB2_1 and TiB2_2 in Tables 3 and 5.

5.       The reviewer didn’t understand how the author divided the core and the solid shell in Fig. 13. Please describe this in detail.

6.       In the carbonization of Mo, (Ti, Mo)C2 and MoC were formed with and without carbon addition, respectively. Please explain these differences and describe the reaction mechanism (or chemical equation) as shown on page 16.

Minor comments

7.       The scale bars in the SEM images are difficult to see, please correct them.

8.       In Figure 1 and 2, there is the same condition of TiB2_0. Please unify its color.

9.       As there are some English mistakes, please check again, e.g., summarised, TiB2 titanium diboride, 2-5 m, etc.

10.   Please unify the words, such as molybdenum disilicide and MoSi2, etc.

11.   The reviewer recommends removing the low magnification SEM images due to the large number of figures.

Comments on the Quality of English Language

There are some mistakes, please check again.

Reviewer 3 Report

Comments and Suggestions for Authors

-line 97-98- please provide more info and formula for KIc according to Niihara formulation

line 99-101- describe principles of determination for Young modulus using the ultrasonic defectoscope, formula.

line 236-241 please provide the pictures with Indents, and crack length measurements.

Line 364- The title is not conclusive, please reformulate it.

improve conclusions at the end

Reviewer 4 Report

Comments and Suggestions for Authors

The manuscript investigates the effects of carbon and MoSi2 additions in the fabrication of TiB2 composites and their mechanical properties. Upon thorough review, I am unable to recommend this manuscript for publication due to several critical issues:

1.       Language Clarity: The language used throughout the manuscript is poor, significantly hindering comprehension. This aspect needs substantial improvement for readability and professionalism.

2.       Lack of Originality and Motivation: The work lacks creativity and fails to establish its motivation clearly. The significance and novelty of the study are not adequately demonstrated, which is essential in scholarly work.

3.       Inadequate Experimental Detail: The manuscript is missing crucial experimental details. For example, the sintering temperature is mentioned as increasing at 10 C/min, but the ultimate temperature is not specified. Another example as following, there is a notable absence of details regarding the SEM images, such as the type of detector used. This omission makes it challenging to compare samples effectively, especially those subjected to different temperatures.

4.       Lack of professionalism in scientific writing: For instance, the incorrect usage of "C" instead of “°C" for temperature is a notable error.

5.       Report-like Presentation of Results: The results are presented more like a report than a scholarly article, lacking in-depth analysis and discussion. Key trends and behaviors, such as linear shrinkage with temperature changes, are not adequately explained.

Comments on the Quality of English Language

TThe language used throughout the manuscript is poor, significantly hindering comprehension. This aspect needs substantial improvement for readability and professionalism.

Round 2

Reviewer 2 Report

Comments and Suggestions for Authors

I am satisfied with the revisions that have been made by the authors.

With respect to response to previous comment 6, the reviewer cannot make a determination as to whether it is reasonable or not. However, it is recommended that it be included as the author's hypothesis, because it will help other researchers. However, the final decision is left to the author.

Author Response

Once again, thank you for your valuable reviewer's comments. We have introduced an additional comment on the content of the article

Reviewer 3 Report

Comments and Suggestions for Authors

for line 108 for the formula use the same principles of the presentation as in formula 2 line 124, with fractions using an equation editor from MS Word.

Author Response

Once again, thank you for your valuable reviewer's comments. We have changed the format of the equation to be consistent with the format of Equation 2.